# Tackling AlfWorld with Action Attention and Common Sense from Pretrained LMs

**Yue Wu**[1], **So Yeon Min**[1], **Yonatan Bisk**[1], **Ruslan Salakhutdinov**[1], **Shrimai Prabhumoye**[1,2]
[1]Carnegie Mellon University, [2]Nvidia Research
{ywu5,soyeonm,ybisk,rsalakhu}@andrew.cmu.edu, sprabhumoye@nvidia.com

## Abstract

Pre-trained language models (LMs) capture strong prior knowledge about the world. This common sense knowledge can be used in control tasks. However, directly generating actions from LMs may result in a reasonable narrative, but not executable by a low level agent. We propose to instead use the knowledge in LMs to simplify the control problem, and assist the low-level actor training. We implement a novel question answering framework to simplify observations and an agent that handles arbitrary roll-out length and action space size based on action attention. On the Alfworld benchmark for indoor instruction following, we achieve a significantly higher success rate ($50\%$ over the baseline) with our novel object masking - action attention method.

## 1 Introduction

Humans can abstractly plan their everyday tasks without execution; for example, given the task "Make breakfast", we can roughly plan to first pick up mug and make coffee, then pick up an egg and scramble it, etc. This capability, if endowed to embodied agents, can help induce generalizable common-sense and reasoning. Recently, a few works Huang et al. (2022a,b); Ahn et al. (2022); Yao et al. (2020) have used large language models (LLM) Bommasani et al. (2021) for abstract planning for embodied or gaming agents. These works have shown incipient success in extracting procedural world knowledge from LLMs in linguistic forms and matching them with executable actions conditioned on the environment.

However, recent works operates in a non-executable setting Huang et al. (2022a) or in domains with limited interactions Huang et al. (2022b); Ahn et al. (2022), primarily just consisting of "moving" objects. In addition, the scenarios considered have been largely simplified from the real world. Ahn et al. (2022) provides all available objects and possible interactions at the start and limits tasks to the set of provided objects/interactions. Huang et al. (2022b) limits the environment to objects on a single table. On the other hand, to successfully "cut some lettuce" in a real-world room, one has to "find a knife.", which can be non-trivial since there can be multiple drawers or cabinets Chaplot et al. (2020); Min et al. (2021); Blukis et al. (2021).

A more realistic scenario leads to long observation, long roll-outs, and large action spaces (Fig 1). Specifically, the text description of the observation increases due to the number of receptacles and objects the agent sees, and the roll-outs accumulated quickly and become too long to fit into any LM. Furthermore, the large and constantly changing action space makes learning difficult.

In this work, we address the three problems with (1) a novel QA framework to filter irrelevant objects (**Object Masking**) and (2) querying long/variable length of actions (**Action Attention**). Object Masking reduces the length of observations, and Action Attention enables the agent to handle arbitrary action space size and roll-out length.

36th Conference on Neural Information Processing Systems (NeurIPS 2022).

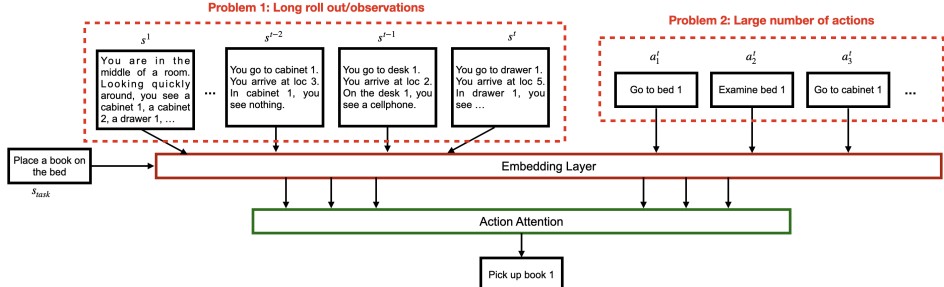

Figure 1: Overview of Action Attention method. Action Attention block is a transformer-based framework that computes a key $k$ for each permissible action and output action scores as dot-product between key and query $q$ from the observations. This method addresses the two problems of: (1) long roll outs and (2) large number of actions.

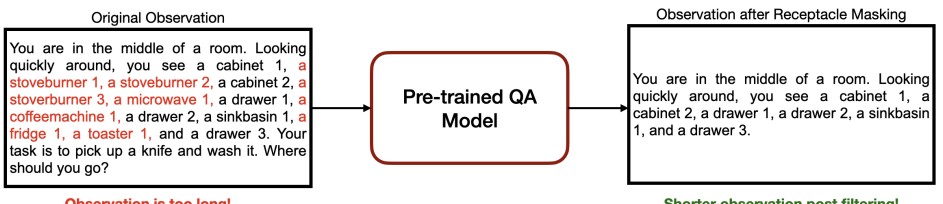

Figure 2: Overview of Receptacle Masking method. We use a pre-trained question answering model to filter irrelevant receptacles/objects in the observation of each scene. As we can see, the original observation is too long and the receptacles shown in red are not relevant for task completion. These receptacles are filtered by the QA model making the observation shorter.

We focus on instruction following in indoor households; on the AlfWorld Shridhar et al. (2020) benchmark. We achieve a significantly higher success rate (absolute $50\%$ over the baseline) with our novel object masking - action attention method. The strong performance of our method demonstrates that large language models can be used as knowledge bases to query common sense for closed-loop intermediate planning.

## 2 Related Work

**Text Games** Text-based games are complex, interactive simulations where the game state and action space are in text. They are fertile ground for language-focused machine learning research. In addition to language understanding, successful play requires skills like memory and planning, exploration (trial and error), and common sense. The AlfWorld Shridhar et al. (2020) simulator extends a common text-based game simulator, TextWorld Côté et al. (2018), to create text-based analogs of each ALFRED scene.

**LMs for Control** LMs have been used for planning high-level policies Huang et al. (2022a); Ahn et al. (2022). Huang et al. (2022a) focus on high-level sub-goals that are not executable directly in most control environment. Ahn et al. (2022) on the other hand, require few-shot demonstrations of up to 17 examples, making the length of prompt infeasible for AlfWorld.

## 3 Methodology

Our method consists of action attention (Fig 1) and receptacle/object masking (Fig 2). The action attention module scores each permissible action with a transformer-based architecture and is trained on imitation learning on the expert. Receptacle/object masking uses a zero-shot QA model to filter out irrelevant objects in the observation.

**Problem Setting** We define the task description as $s_{\text{task}}$, the observation string at time step $t$ as $s^t$, the list of permissible actions $\{a_i^t | a_i^t \text{ can be executed}\}$ as $A^t$. For each observation string $s^t$, we define the receptacles and objects within the observation as $r_i^t$ and $o_i^t$ respectively. We are interested in learning a policy $\pi$ that outputs the optimal action among permissible actions.

**Action Attention**   Since the number of permissible actions can vary a lot by environment, the agent needs to handle arbitrary dimensions of action space. While Shridhar et al. (2020) addresses this challenge by generating actions token-by-token, such generation process leads to degenerate performance even on the training set.

We eschew the long roll out/ large action space problems by (1) representing observations by averaging over history, and (2) individually encoding actions (Fig 1). In our proposed action attention framework, we first represent historical observations $H^t$ as the average of embeddings of all individual observations through history, and $H^A$ as the list of embeddings of all the current permissible actions (Eq. 1). Then, in Eq. 2, we compute the query $Q$ using a transformer with a "query" head ($\mathcal{M}_{\mathcal{Q}}$) on task embedding ($H^t$), the current observation embedding ($s^t$), and the list of action embeddings ($H^A$). In Eq. 3 we compute the key $K_i$ for each action $a_i$ using the same transformer with a "key" head ($\mathcal{M}_{\mathcal{K}}$) on task embedding ($H^t$), the current observation embedding ($s^t$), and embedding of action ($a_i$).

Finally, we compute the dot-product of the query and keys as action scores for the policy $\pi$ (Eq. 4).

$$H^t = \text{avg}_{j \in [1, t-1]} \text{Embed}(s^j), \; H^A = \left[\text{Embed}(a_1^t), ..., \text{Embed}(a_n^t)\right] \tag{1}$$

$$Q = \mathcal{M}_{\mathcal{Q}} \left(\text{Embed}(s_{\text{task}}), H^t, \text{Embed}(s^t), H^A\right) \tag{2}$$

$$K_i = \mathcal{M}_{\mathcal{K}} \left(\text{Embed}(s_{\text{task}}), H^t, \text{Embed}(s^t), \text{Embed}(a_i^t)\right) \tag{3}$$

$$\pi = \text{softmax} \left([Q \cdot K_i | i \in \text{all permissible actions}]\right) \tag{4}$$

**Receptacle/Object Masking**   Typical Alfworld scenes can start with around 15 receptacles, each containing up to 15 objects. An agent starting with no knowledge about where to look for objects that are relevant to solving the task at hand can easily get stuck. We make the observation that many receptacles and objects are irrelevant to specific tasks during both training and evaluation, and can be easily filtered with common-sense about the tasks. For example, in Fig 2 the task is to pick up and wash a knife. By removing the irrelevant receptacles like the toaster, fridge, stoveburners, we could significantly shorten our observation.

We propose to leverage commonsense knowledge captured by large pre-trained QA models. Note that we do not fine-tune the pre-trained QA model for our particular task but we use it in a zero-shot manner. We create prompt in the format "Your task is to: <task string>. Where should you go to?" for receptacles and "Your task is to: <task string>. Which objects will be relevant?" for objects. We then obtain scores from the pre-trained QA model representing whether the model believe that the receptacle/object is relevant, and we mask out irrelevant receptacles/objects that have scores below a threshold.

## 4   Experiments and Results

**Hyper-parameters.**   For the common-sense language model we choose Macaw-11b Tafjord and Clark (2021), which is reported to have common sense QA performance on par with GPT3 Brown et al. (2020) while being order of magnitudes smaller. For embedding of actions and observations, we use pretrained RoBERTa-large Liu et al. (2019) with embedding dimension 1024. Our transformer ($\mathcal{M}_{\mathcal{Q}}$ and $\mathcal{M}_{\mathcal{K}}$) is a 12-layer transformer with 12 heads and hidden dimension 768. The last layer is then fed into two linear heads to generate $K$ and $Q$. For receptacle/object masking, we use a score threshold of $0.4$ below which the objects are masked out.

**Baselines.**   We use the BUTLER::BRAIN (**BUTLER+CG**) agent presented in Shridhar et al. (2020), which consists of an encoder, an aggregator, and a decoder. At each time step $t$, the encoder takes initial observation $s^0$, current observation $s^t$, and task string $s_{\text{task}}$ and generates representation $r^t$. The recurrent aggregator combines $r^t$ with last recurrent state $h^{t-1}$ to produce $h^t$, which is then decoded into a string $a^t$ representing action. In addition, the BUTLER agent uses beam search to get out of stucked conditions in the event of failed action. Our second baseline (**BUTLER+AC**) is an implementation by Shridhar et al. (2020) to allow BUTLER to directly choose from admissible commands. Both BUTLER agents are trained with an online imitation learning curriculum, DAgger Ross et al. (2011), assisted by a rule-based expert.

| Model | DAgger | | Behavior Cloning | | |
|---|---|---|---|---|---|
| | seen | unseen | train | seen | unseen |
| BUTLER + CG Shridhar et al. (2020) | 40 | 35 | 9 | 10 | 9 |
| BUTLER + AC* Shridhar et al. (2020) | 61.7 | 16.89 | - | - | - |
| Action Attention | $90.41 \pm 0.02$ | $33.42 \pm 0.05$ | 30 | 25 | 9 |
| Action Attention + Masking | $\mathbf{90.53} \pm 0.02$ | $\mathbf{34.92} \pm 0.03$ | **30** | **25** | **11** |

Table 1: Average completion rate with DAgger and Behavior Cloning. **\*** Shridhar et al. (2020) did not provide evaluation for BUTLER+AC, so we report the performance from our own experiment.

## 4.1 Results

The results of both DAgger and Behavior Cloning are shown in Table 1. We observe that both the baselines and our models benefit greatly from DAgger training. However, DAgger assumes an expert that is well-defined at all observation spaces, which is infeasible in most practical scenarios. We also observe that training is 100x slower with DAgger compared to behavior cloning (3 weeks for DAgger v.s. 6 hours for Behavior Cloning).

In the DAgger training scenario, our action attention agent greatly exceeds baseline performance in **seen** evaluation (we observe a $50.41\%$ absolute improvement), and receptacle/object masking further improves the performance on unseen evaluation.

In the behavior cloning scenario, where there is not enough training data, we observe that Receptacle/Object Masking is more effective in the behavior cloning setting (we observe a $22.2\%$ relative improvement).

**Quality of Pre-trained QA for receptacle/object masking**  We evaluate the zero-shot receptacle/object masking performance of Macaw on the three splits of AlfWorld. In Fig 3, we plot the AUC curve of the relevance-score that the model assigns to the objects v.s. objects that the rule-based expert interacted with when completing each task. In practice decision threshold of $0.4$ retains around $80\%$ relevant objects, $70\%$ relevant receptacles and reduces the length of observations by $50\%$ on average. In addition, the zero-shot QA model demonstrates consistent masking performance on all three splits of the environment, even on the unseen split.

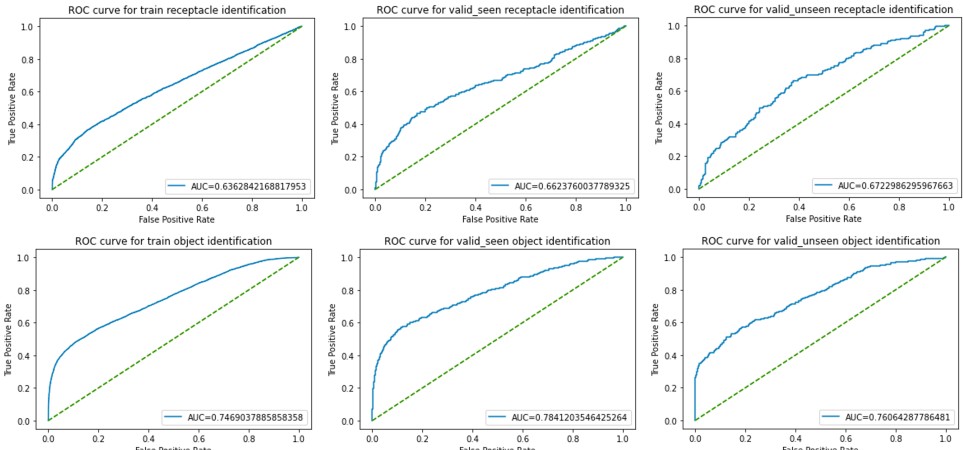

Figure 3: Plot of AUC scores of zero-shot relevance identification across all tasks in the Alfworld-Thor environment, with the Macaw-11b model. The ground truth is obtained as receptacles/objects accessed by the rule-based expert. **Top:** Receptacle relevance identification. **Bottom:** Object relevance identification. The QA model achieves average AUC-ROC score of 65 for receptacles, and 76 on objects.

## 5 Conclusion

In this work, we present (1) a novel question answering framework to simplify observation and (2) an action attention framework to handle large and variable size action space. Future works can focus on adding ways from which LMs can assist learning of the policy, such as providing high-level plans.

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
