# OpenReview forum: "Tackling AlfWorld with Action Attention and Common Sense from Language Models"
_NeurIPS.cc/2022/Workshop/LaReL — LaReL 2022_

### Official Review · Reviewer_a1ex · 2022-10-16
**Good improvement over baseline in some cases, but presentation should be reworked.**

**Rating:** 6
**Confidence:** 2

**Review:**

The authors introduce a new scheme to address the long rollouts and large action spaces seen in some closed-loop control problems. The scheme has two parts: 1. using a language model (LM) to reduce the action space of an agent by removing actions that would violate common-sense (object masking) and 2. using an attention mechanism to score all the actions in a large, variable-sized action space. The resulting scheme achieves a 30% absolute improvement over a BUTLER baseline on AlfWorld (61.7% -> 90.5%) in the “seen” Dagger evaluation.

The work is in general presented in a clear and legible manner, with some exceptions I highlight below.

Of the two parts of the author’s scheme, nearly all uplift over the baseline is from the Action Attention. Including the external language model to reduce the size of the action space has very little additional uplift (if any) over pure Action Attention. I highlight this as a large fraction of the paper’s exposition focusses on the use of the external LM. I feel the paper would be more impactful if the authors devoted more exposition to ablations and discussion of the Action Attention, which is in my view the main contribution of the paper.

As a further note on presentation, I felt equations 1 through 4, which explain their use of the transformer, to be very unclear.
The use of the \mathcal{M} symbol to denote both the query and key operation is confusing. If the authors are indeed using an identical neural network to generate both keys and queries then this should be clarified as it is a very non-standard use of a transformer.
Furthermore, equations 2 and 3 are formatted poorly. They have differing numbers of inputs and use of brackets. As a reader, I am left to guess that the square brackets represent a form of concatenation? Sentences such as “we compute the query Q with a transformer (\mathcal{M}) on the task embedding” are confusing. It would be clearer if the authors elaborate on exactly how the transformer was used. For instance, is Q the activations before the final reshaping layer or is it taken from an intermediate query layer in the transformer? I fear it would be hard to for someone else to replicate this work without a rewrite of section 3, potentially with the inclusion of an additional diagram.

I would also liked to have seen the author’s more strongly motivate their choice of baseline. If the baseline is the currently SOTA then it should be mentioned explicitly. If the baseline is non-SOTA then I believe the SOTA result should be included or some minimal rational given for its exclusion.

Pros:
Good improvements over baseline in various scenarios.
Presentation is generally clear.
The idea of using an external LM for common-sense reasoning for this particular task is interesting and, as far as my knowledge goes, novel.
The action attention mechanism to handle large and variable action spaces seems powerful (but I am less confident of its novelty).

Cons:
I believe the paper should be restructured to focus more on action attention, particularly ablating the various design decisions (e.g. averaging the embeddings of historical observations).
The explanation of the author’s use of a transformer was unclear and should be rewritten.
The author’s should include more motivation for their choice of baseline.

---

### Official Review · Reviewer_CGUa · 2022-10-20
**Improvements over Alfworld baseline, need to improve clarity**

**Rating:** 6
**Confidence:** 3

**Review:**

The authors improve over the Alfworld benchmark (instruction following domain with text state and action space) via the following two contributions: a) actions are selected via particular action attention block, b) shortening of environment description by filtering out instruction-irrelevant containers through the use of pre-trained QA models.

The topic is highly relevant for this workshop and it's good to see more work exploring the use of pre-trained models in RL tasks. However, the use of attention in action selection is not particularly novel, it would be great to clarify the exact novelty of the approach by highlighting the differences between the proposed action attention block and the existing work. The use of QA model is specific to this domain (the choice of which receptacle to open next), do you have thoughts on how this approach and queries used for shortening the observation could be made more general? Moreover, the improvement seen from adding the masking seems very small to negligible (Table 1), why is that? More experiments illuminating the importance of masking would make the paper stronger.

The following are suggestions for improving the clarity of the paper:
- in Introduction, please define better what is meant by open and closed-loop planning; also doesn't Huang et al 2022 enable closed-loop planning?
- there is no reference to AlfWorld in the first couple of times it is mentioned
- in Related Work, you refer to the most relevant papers, however, this whole section should be longer (& could be moved in the Appendix if needed) and contrast better your contribution to the existing work

---

### Decision · Program_Chairs · 2022-10-21

Accept